# Association of IL13 polymorphisms with susceptibility to myocardial infarction: A case-control study in Chinese population

**Rong Chen[1], Qiaoling Bao[2], Xiaofeng Ma**[1]*

**1** Department of Cardiology, Qinghai Province Cardiovascular and Cerebrovascular Disease Specialist Hospital, Xining, Qinghai, China, **2** Department of Coronary Heart Disease, Qinghai Province Cardiovascular and Cerebrovascular Disease Specialist Hospital, Xining, Qinghai, China

* Xiaofeng_Ma_cardio@outlook.com

## Abstract

**Data Availability Statement:** All minimal data are in the manuscript. if further info is required, request can be directed to the corresponding author.

**Funding:** The author(s) received no specific funding for this work.

### Background

Inflammatory cytokines play a major role in the pathogenesis of myocardial infarction (MI). Although information on the importance of interleukin 13 (IL13) in human MI is limited, it has been well documented in the mouse model. Genetic variation in the IL13 gene has been associated with the structure and expression of the IL13. In the present study, we hypothesized that IL13 common genetic variants would be associated with a predisposition to the development of MI.

### Materials and methods

The present study enrolled 305 MI patients and 310 matched healthy controls. Common genetic polymorphisms in the IL13 gene (rs20541, rs1881457, and rs1800925) were genotyped using the TaqMan SNP genotyping method. Plasma levels of IL13 were measured using an enzyme-linked immunosorbent assay (ELISA).

### Results

In MI patients, minor alleles of the IL13 rs1881457 and rs1800925 polymorphisms were less common than in healthy controls [rs1881457: AC (P = 0.004, OR = 0.61), C (P = 0.001, OR = 0.66); rs1800925: CT (P = 0.006, OR = 0.59)]. Further haplotype analysis of three studied SNPs revealed a significant association with predisposition to MI. Interestingly, IL13 rs1881457 and rs1800925 were linked to plasma levels of IL13: the reference genotype had higher levels, heterozygotes were intermediate, and the alternate genotype had the lowest levels.

### Conclusions

In the Chinese population, IL13 (rs1881457 and rs180092) variants are associated with different plasma IL13 levels and offer protection against MI development. However, additional

**Competing interests:** The authors have declared that no competing interests exist.

**Abbreviations:** MI, Myocardial infarction; SNP, Single nucleotide polymorphism; HWE, Hardy-Weinberg equilibrium; ELISA, Enzyme-linked immunosorbent assay; IL, interleukin; TNF, tumor necrosis factor; STEMI, ST-Elevation Myocardial Infarction; NSTEMI, Non-ST-Elevation Myocardial Infarction; ECG, Electrocardiogram.

research is required to validate our findings in different populations, including descent samples.

## Introduction

Myocardial infarction (MI) is a serious form of coronary artery disease (CAD) that results from the occlusion of a coronary artery, impairing blood supply to the heart muscle. Despite significant advances in the understanding and treatment of cardiovascular diseases, MI remains a major public health concern worldwide. A recent meta-analysis revealed the global prevalence of MI is around 3.8% below 60 years old individual and the prevalence is 9.5% in older age group (>60 years) [1]. The mortality rate has increased in Chinese populations both in rural and urban areas in between 2002 to 2016 [2]. MI can be categorized as either STEMI or NSTEMI based on ECG changes. STEMI involves a complete obstruction of a coronary artery, which is indicated by ST-segment elevation on an ECG, necessitating immediate reperfusion therapy. On the other hand, NSTEMI results from a partial blockage and presents without ST elevation but with other ECG changes, which are managed with medications and, in some cases, percutaneous coronary intervention. In the context of MI, STEMI is more commonly observed than NSTEMI [3]. Both types of MI require prompt medical attention to prevent heart damage and improve patient outcomes. The multifaceted aetiology o MI includes complex interactions between genetic predisposition and environmental factors [4]. Numerous environmental factors and host genetic factors have been demonstrated to impact cytokines and the immune response in humans [5]. Cytokines are critical in the pathophysiology of MI, as they mediate the inflammatory response that follows cardiac tissue injury [6]. These small, signaling proteins are secreted by immune cells, endothelial cells, and other cell types in the heart, orchestrating a complex series of events aimed at limiting damage and initiating repair [7]. During an MI, cytokines such as interleukins, tumor necrosis factor-alpha (TNF-$\alpha$), and chemokines are rapidly increased, contributing to both protective and detrimental effects [8] and facilitate leukocyte infiltration, modulate cell survival and apoptosis, and influence myocardial remodelling [9]. A recent study found differences in inflammatory molecules between ST elevation myocardial infraction (STEMI) and non STEMI clinical phenotypes, which are distinguished by elevated methylene diphosphonate (MDP), macrophage inflammatory protein-1β (MIP-1β), and TNF-$\alpha$ [6]. Additionally, cytokine levels were related to blood flow through the infract related artery [6]. Interleukin-13 (IL-13), is a pro-fibrotic cytokine that is secreted by various immune cells, including T cells, B cells, and macrophages. This cytokine plays a crucial role in the Th2 immune response, a process that is triggered by the activation of T helper 2 cells. These cells release a range of cytokines, such as IL-4, IL-5, and IL-13, which promote B cell activation, class switching to IgE production, and the recruitment of eosinophils [10]. Although IL-13 was initially thought to play a role in allergic responses, it has gained increasing attention due to its involvement in a variety of immune processes and diseases. The cytokine exerts multiple and diverse biological effects on different cell types or tissues and is responsible for immune cell differentiation, proliferation, and inflammatory responses [10]. In the MI mouse model, dynamic expression of IL13 has been reported; after the incidence of MI, the levels significantly increased, reaching a peak on day three and then declining until day seven, when they increased again [11]. However, another research group found that upregulation began on day 7 and continued until day fourteen [12]. Surprisingly, investigations on the role of IL13 in MI patients are very limited and in a study lower levels have been reported [13].

Genetic variations can influence gene expression, protein function, and susceptibility to different disease risks [14]. Functional variations in the IL13 gene have been linked with different types of cardiovascular diseases [15]. Although various genome-wide association studies have been carried out in diverse populations, no notable genetic association between the IL13 gene and susceptibility to develop MI has been identified [16, 17]. Given that differential IL13 levels have been observed in MI patients and functional genetic variants play a role in regulating IL13 levels, we hypothesized that common genetic variants affecting plasma IL13 levels would be linked to the pathogenesis of MI in the Chinese population. Till date several single nucleotide polymorphisms (SNP) in IL13 gene have been reported to be linked with wide range of clinical diseases (https://www.ncbi.nlm.nih.gov/clinvar/?term=IL13[gene]). However, three SNPs (rs20541, rs1881457 and rs1800925) have been studied most frequently based on their functional relevance. The human IL13 gene, which consists of 5 exons and 4 introns, is located in the long arm of the fifth chromosome (q31.1). The IL13 rs20541 polymorphism is found in the fourth exon, which is a result of a transition mutation (A>G) that alters the amino acid sequence of the IL13 protein at the 130[th] codon (Glycine to Arginine). The other two genetic variants are situated in the promoter region of IL13 gene (rs1881457: A-1512C; rs1800925: C-1111T) and are presumed to alter the expression of IL13 [18]. In addition, the significance of IL13 polymorphisms has been documented in clinical phenotypes that serve as risk factors for MI, such as type-1 diabetes mellitus, hypertension, and dyslipidemia. Specifically, in Kuwaiti children, the rs20541 variant was associated with susceptibility to T1DM [19]. Conversely, genetic variants in the IL13 gene failed to demonstrate an association with T1DM in Filipino [20], British [21], and Thai populations [22]. Moreover, the rs1800925 and rs1881457 polymorphisms have been shown to be linked to hypertension in the Korean population [23]. Based on the functional significance of these genetic polymorphisms, case-control studies in various populations have been conducted to investigate their association with MI susceptibility. In the Iranian population, rs1881457 heterozygotes provided protection against the development of MI, but no significant link between MI susceptibility and the rs20541 polymorphism was found [24]. In contrast, the IL13 rs20541 variant increased the risk of MI in Greek Cypriot males [25].

Given the importance of IL13 in the pathogenesis of MI and the possibility of genetic variants influencing IL13 levels, we hypothesized that the common IL13 genetic variant would be associated with susceptibility/resistance to MI. We conducted a hospital-based case-control study in the Chinese population to investigate the possibility of an association between IL13 variants and MI.

## Materials and methods

### Study design

The study used a hospital-based case-control design to investigate the genetic association between common variations of the IL13 gene and the susceptibility or resistance to MI in a Chinese cohort. The report of this study adhered to the STREGA guidelines.

### Subjects

The present study was conducted on the patients enrolled in the Department of Cardiology, Qinghai Province Cardiovascular and Cerebrovascular Disease Specialist Hospital, from January 2017 to December 2022. Other baseline parameters such as hypertension, hypercholesterolemia, hyperglycemia, and coronary artery disease were also explored in the included MI patients. Age, sex and gender-matched healthy controls hailing from similar geographical locations were considered healthy controls without any history of heart-related issues. All healthy controls were subjected to blood pressure measurement and different biochemical tests such as cholesterol and blood sugar levels, and subjects with higher BP (more than 120 mm Hg for

systolic and 80 mm Hg for diastolic), cholesterol (total >200 mg/dL, LDL >100 mg/dL, HDL<40 mg/dL for eman and 50 mg/dL for women, triglycerides > 150 mg/dL), or sugar levels (fasting blood glucose > 99 mg/dL and HbA1c >5.7%) were not included in the present study. The study protocol was approved by the institutional ethical committee of Qinghai Province Cardiovascular and Cerebrovascular Disease Specialist Hospital, and informed written consent was obtained from each participant. In situations where patients were unable to provide informed consent due to their medical condition, consent was obtained from legally authorized representatives, including family members or legal guardians. The IEC reviewed and approved the study protocol, including this consent process (IEC/2016/156). If patients subsequently regained capacity, they were informed and requested to provide direct consent. About 5 ml of intravenous blood was collected from MI (within six hours of the MI onset) and healthy control subjects with anticoagulants.

## Sample size calculation

The sample size for the study was calculated a priori to ensure that an adequate number of cases and controls were enrolled. The GPower v3.1.9.7 software was utilized for the sample size calculation. To achieve a power of 90% with an effect size of 0.15 and an alpha error probability of 0.05, the analysis indicated that 630 subjects were needed. Consequently, the study was designed to include 315 cases and an equal number of healthy controls in the investigation of the genetic association.

## IL13 genotyping

Whole genomic DNA was isolated from the whole blood by QIAamp Blood mini kit according to the manufacturer's instructions (QIAGEN, Germany). DNA quantity and quality were assessed using a NanoDrop 2000 spectrophotometer (Thermo Fisher Scientific, Waltham, MA, USA) and agarose gel electrophoresis. Genotyping of common IL13 gene polymorphisms (rs20541, rs1881457 and rs1800925) were carried out by TaqMan Genotyping assays kit (Thermo Fisher Scientific, Catalog number: 4351379) (https://tools.thermofisher.com/content/sfs/manuals/TaqMan_SNP_Genotyping_Assays_man.pdf). Details of the probe are given below: rs20541, C___2259921_20, Context Sequence [VIC/FAM] `TTAAAGAAACTTTTTCGCGAGGGAC[A/G]GTTCAACTGAAACTTCGAAAGCATC`, rs1881457: C__11740467_10, Context Sequence [VIC/FAM]`TACAGATTAGGAAACAGGCCCGTAG[A/C]GGGGTCACACGGCCAAGTAGCGGCA`, rs1800925, C___8932056_10, Context Sequence [VIC/FAM]`GGTTTCTGGAGGACTTCTAGGAAAA[C/T]GAGGGAAGAGCAGGAAAAGGCGACA`. The genotyping reactions comprised the genomic DNA template, TaqMan Genotyping Master Mix, and the specific primer/probe combination. These reactions were performed using a real-time PCR thermal cycler, with an initial denaturation step at 95˚C for 10 minutes, followed by 40 cycles of denaturation at 95˚C for 15 seconds and annealing/extension at 60˚C for 1 minute. The fluorescence signals were detected and analyzed. To ensure the accuracy and reliability of the genotyping results, positive and negative controls were included in each run. Genotype calling was accomplished based on the allelic discrimination plot. Each sample was considered for genotyping in duplicate and considered for analysis only in case of concordant observation. Subjects who were unsuccessful in genotyping were excluded from the present investigation.

## Plasma IL13 quantification

Plasma levels of IL-13 were measured by enzyme-linked immunosorbent assays (ELISA) according to the manufacturer's instructions (Invitrogen, Catalog no: BMS231-3, https://www.thermofisher.com/document-connect/document-connect.html?url=https://assets.thermofisher.com/TFS-Assets%2FLSG%2Fmanuals%2FMAN0016602_231–3_HuIL-13ELISA_UG.pdf). The

plasma samples, which were collected and stored at -80˚C until analysis, were thawed on ice before use. To prepare a standard curve, the standards were reconstituted with distilled water and serially diluted (1:2) on seven tubes, such that the highest concentration of the standard remained at 100 pg/mL and the lowest was 1.6 pg/mL. The precoated microplate wells were properly washed, and the pre-prepared standards were applied in duplicate wells. Two wells were used as blank wells and were only filled with assay buffer. 50 microliters of assay buffer were applied to all sample wells, and an equal amount of samples were applied to the wells in duplicate. Fifty microliters of conjugate mixture were added to all wells, and the plate was incubated at room temperature for two hours. After incubation, the plate was washed thrice properly and proceeded to the color development step using TMB as a substrate solution. The reaction stopped ten minutes later, and absorbance was measured at 450 nm. The concentrations of IL-13 in the samples were determined by interpolation from the standard curve.

## Statistical analysis

All statistical analysis was performed by GraphPad Prism v9 (GraphPad Software, Boston) using the default setting. The allele and genotype frequency was calculated by manual counting. The distribution of genotypes with references to Hardy Weinberg equilibrium (HWE) was explored using the Microsoft Excel program. The Fisher exact test compared the prevalence of genotype and allele in different clinical categories. The Fisher exact test is a reliable method for obtaining precise test results in small sample sizes. It calculates the exact probability value and the confidence interval of the observed data for measuring the association [26]. Bonferroni correction is a technique used to correct the significance level for multiple comparisons, aiming to reduce the likelihood of false positives. It entails dividing the desired significance level ($\alpha$) by the number of comparisons (n), thereby creating a more stringent threshold ($\alpha$/n) [27]. In the current analysis, three SNPs were examined, and the significance levels were adjusted using the Bonferroni correction method. A P value less than 0.016 (0.05 divided by 3) was considered statistically significant. The haplotype construction and comparison of their distribution among MI patients and healthy controls were executed by SNPAlyze software Version 8.1.1 employing the default setting (DYNACOM Co. Ltd. Japan). The software uses a permutation test for the comparison of haplotype frequencies among two groups. The mean plasma IL13 levels in healthy controls and MI patients were compared by student's t-test. The association of IL13 polymorphism with the plasma levels of IL13 was explored by one-way analysis of variances (ANOVA). For the comparison of cytokine among cases and controls or within different genotypes of IL13 polymorphisms, a P value less than 0.05 was considered statistically significant.

## Results

### Baseline characteristics of patients and controls

The baseline characteristics of patients and controls are shown in Table 1. In the present study, a total of 305 MI patients (Male: 189, Female: 116) were enrolled. In addition, 310 age and gender-matched (male: 195, female: 115) healthy controls were considered. The mean age of patients was 56.65 years, and healthy controls were 55.35 (p = 0.31). Hypercholesterolemia (58%) was more frequent among MI patients, followed by hyperglycemia (53%) and hypertension (52%). About one-third of the total MI patients had coronary artery diseases. Interestingly, the family history of cardiovascular diseases or smoking habit was more frequent in the MI patients compared to the healthy controls. While the excessive alcohol consumption rate was comparable among MI patients and controls.

**Table 1. Baseline characteristics of myocardial infarction patients and healthy controls.**

| Characteristics | Myocardial infarction (n = 305) | Healthy controls (n = 310) | P value |
|---|---|---|---|
| Gender (male: female) | 189: 116 | 195: 115 | 0.86* |
| Age (mean ± SD) | 56.65 ± 18.34 | 55.35 ± 13.54 | 0.31& |
| Hypertensions | 159 (52%) | -- | |
| Hypercholesterolemia | 177 (58%) | -- | |
| Hyperglycemia | 162 (53%) | -- | |
| Coronary artery disease | 95 (31%) | -- | |
| High-density lipoprotein (mean ± SD) | 35.66 ± 8.42 | 52.64 ± 11.17 | <0.0001& |
| Low-density lipoprotein (mean ± SD) | 168.54 ± 28.83 | 102.53 ± 15.37 | <0.0001& |
| Triglycerides (mean ± SD) | 191.49 ± 31.36 | 125.1 ± 22.94 | <0.0001& |
| Total cholesterol (mean ± SD) | 241.24±36.26 | 184.41±24.69 | <0.0001& |
| Family history of cardiovascular disease | 128 (42%) | 71 (23%) | <0.0001* |
| Smoking habit | 178 (58%) | 128 (41%) | <0.0001* |
| Excessive Alcohol consumption | 134 (44%) | 119 (38%) | 0.16* |

Note: Data are shown in number (%), unless and otherwise specified. SD: standard deviation. Compared by

*Chi square test and

& students t test.

## MI patients displayed higher levels of IL-13 compared to healthy controls

Based on the availability of plasma samples, a total of 224 plasma samples were quantified for plasma levels of IL13, including each of 112 patients and controls, by ELISA. As shown in Fig 1, the MI patients displayed significantly higher mean levels of IL13 compared to healthy controls (p<0.0001).

## Distribution of IL13 polymorphisms in healthy controls

In the present investigation, we genotyped three common gene polymorphisms in the IL13 gene (rs20541, rs1881457, and rs1800925) by TaqMan genotyping assays. Although we have enrolled 328 MI patients and 322 healthy controls for the present study, 310 cases and 305 healthy controls were successfully genotyped for IL13 polymorphisms. The prevalence of these polymorphisms is demonstrated in Table 2. GG genotype (56%) of rs20541 polymorphism was more frequent than heterozygotes (34%) and AA type (10%). The genotype distributions did not match HWE ($\chi^2$ = 6.03, p = 0.01). For the other two gene polymorphisms (rs1881457 and rs1800925), the reference genotypes (AA: 47%, CC: 63%) were more frequent than heterozygotes (AC: 44% and CT: 32%) and alternative genotypes (CC: 7% and TT: 5%), respectively. The distribution of genotypes for rs1881457 ($\chi^2$ = 0.21, p = 0.64) and rs1800925 ($\chi^2$ = 0.54, p = 0.46) polymorphisms were in line with the HWE. The HWE holds substantial implications in population genetics, serving as a foundational model for comprehending the genetic composition of populations. It posits that allele and genotype frequencies in a population remain constant from generation to generation in the absence of evolutionary influences, such as selection, mutation, migration, genetic drift, and non-random mating. HWE offers a null hypothesis for detecting evolutionary forces acting on a population, and any departure from it can indicate the presence of these forces, thereby providing insights into the evolutionary dynamics at play [28, 29].

## IL13 polymorphisms (rs1881457 and rs1800925) are associated with MI

The distribution of IL13 polymorphisms (rs20541, rs1881457, and rs1800925) was explored in MI patients and healthy controls. Details are depicted in Table 2. Genotypes and allele

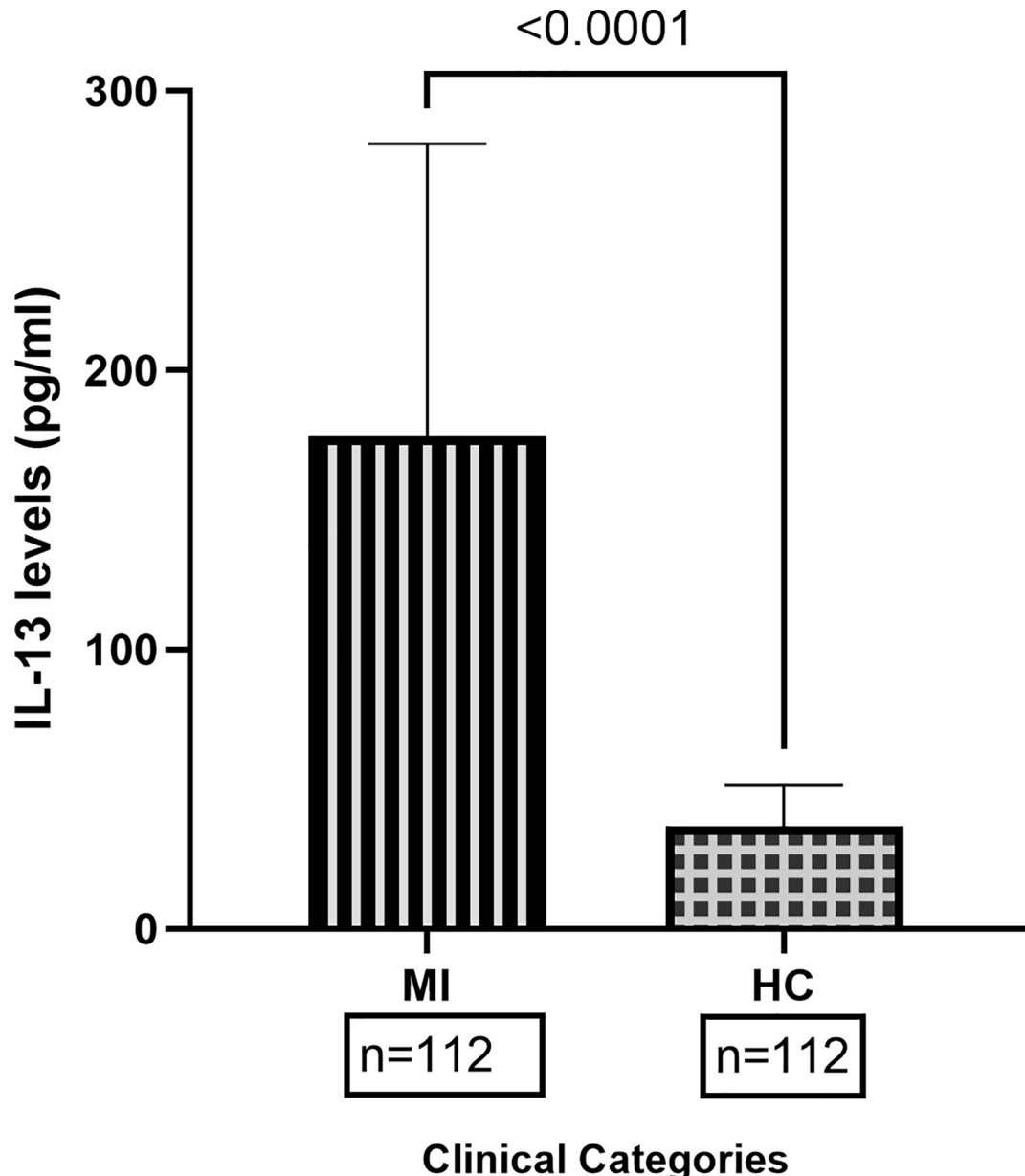

**Fig 1. Plasma levels of IL13 in myocardial infarction patients and healthy controls.** Levels of IL-13 were quantified by ELISA in plasma of MI patients (n = 112) and healthy controls (n = 112). The mean IL13 levels were compared by student's t test. The MI patients displayed higher levels of IL13 compared to the healthy controls. A P value less than 0.05 considered as significant.

distributions were comparable in patients and controls for IL13 rs20541 polymorphism. However, the heterozygotes of rs1881457 and rs1800925 polymorphisms were more frequent in healthy controls than the MI patients (rs1881457: p = 0.004, OR = 0.61; rs1800925: p = 0.006, OR = 0.59). Similarly, the alternate allele of rs1881457 polymorphism (C) was also highly prevalent in healthy controls compared to MI patients (p = 0.001, OR = 0.66), indicating the alternate allele's protective nature against the development of MI in the Chinese population. However, the prevalence of the alternate genotypes for rs1881457 and rs1800925 polymorphisms was comparable among MI patients and healthy controls.

**Table 2. Distribution of IL-13 polymorphisms in myocardial infarction patients and healthy controls.**

| IL-13 Polymorphisms | Genotype or Allele | Healthy Controls (n = 310) | Myocardial Infarction patients (n = 305) | P value | OR (95% CI) |
|---|---|---|---|---|---|
| rs20541 | Genotype | | | | |
| | AA | 31 (10) | 34 (11) | | |
| | AG | 105 (34) | 110 (36) | 0.88 | 0.95 (0.53 to 1.68) |
| | GG | 174 (56) | 161 (53) | 0.58 | 0.84 (0.49 to 1.41) |
| | Allele | | | | |
| | A | 167 (27) | 178 (29) | | |
| | G | 453 (73) | 432 (71) | 0.40 | 0.89 (0.69 to 1.14) |
| rs1881457 | Genotype | | | | |
| | AA | 146 (47) | 183 (60) | | |
| | AC | 136 (44) | 104 (34) | **0.004** | **0.61 (0.43 to 0.85)** |
| | CC | 28 (7) | 18 (6) | 0.04 | 0.51 (0.27 to 0.94) |
| | Allele | | | | |
| | A | 428 (69) | 470 (77) | | |
| | C | 192 (31) | 140 (23) | **0.001** | **0.66 (0.51 to 0.85)** |
| rs1800925 | Genotype | | | | |
| | CC | 195 (63) | 220 (72) | | |
| | CT | 99 (32) | 67 (22) | **0.006** | **0.59 (0.41 to 0.85)** |
| | TT | 16 (5) | 18 (6) | 1 | 0.99 (0.48 to 1.95) |
| | Allele | | | | |
| | C | 489 (79) | 507 (83) | | |
| | T | 131 (21) | 103 (17) | 0.05 | 0.75 (0.57 to 1.01) |

Note. Data are no. (%) of participants unless otherwise specified. The genotype and allele prevalence in MI patients and healthy controls were compared by Fisher's exact test. Wildtype or major allele was taken as reference genotype. A P value less than 0.016 is considered and statistically significant after the Bonferroni correction of the significance level. OR, odds ratio; CI, confidence interval.

### Association of rs20541, rs1881457, and rs180092 haplotype with MI

Haplotype frequency was calculated and compared among the MI patients and healthy controls by SNPAlyze software, and the results are shown in Table 3. Haplotype rs20541-rs1881457-rs1800925, G-A-C (p = 0.02), and A-A-C (p = 0.01) were more frequent in MI patients than healthy controls. In contrast, the haplotypes G-C-C (p = 1E-3), A-C-C (p = 9E-3), G-A-T (p = 0), and A-A-T (p = 2E-3) were more prevalent in controls than in MI patients.

**Table 3. Haplotype distribution of IL13 polymorphisms in MI patients and healthy controls.**

| Haplotype rs20541-rs1881457-rs1800925 | Healthy controls (n = 310) | MI patients (n = 305) | P value |
|---|---|---|---|
| G-A-C | 0.49 | 0.56 | 0.02 |
| A-A-C | 0.13 | 0.19 | 0.01 |
| G-C-T | 0.07 | 0.08 | 0.51 |
| G-C-C | 0.11 | 0.04 | 1E-3 |
| A-C-T | 0.06 | 0.07 | 0.63 |
| A-C-C | 0.05 | 0.01 | 9E-3 |
| G-A-T | 0.04 | 3.534E-3 | 0 |
| A-A-T | 0.01 | 1.459E-9 | 2E-3 |

Note: the haplotype frequency was calculated and compared among healthy controls and MI patients by SNPAlyze software v8.1.1. frequencies of different haplotypes are shown in percentage.

## Plasma levels of IL13 linked with IL13 polymorphisms

To explore the functional relevance of the studied SNPs in the IL13 gene, plasma levels of IL13 were quantified by ELISA, and possible association with different genotypes of IL13 was explored. As shown in Fig 2, the rs1881457 and rs1800925 polymorphisms were linked with plasma levels of IL13. The wildtype of rs1881457 and rs1800925 polymorphisms displayed significantly higher levels of IL13 compared to their respective heterozygotes and homozygous mutants. Heterozygotes had intermediate levels of IL13 (Fig 2A and 2B). Interestingly, a similar pattern of association between IL13 polymorphisms and plasma levels of IL13 remained valid still after separating MI patients (Fig 2C and 2D) and healthy controls (Fig 2E and 2F)

## Discussions

The current hospital-based case-control study found a significant link between IL13 common gene polymorphisms (rs1881457 and rs180092) and MI risk. Furthermore, a substantial relationship was found between plasma IL13 levels and IL13 promoter variants (rs1881457 and rs180092). To the best of our knowledge, this is the first study to investigate the possible association of IL13 polymorphism with the development of MI in the Chinese population.

The role of IL13 in MI has been widely investigated in experimental model systems, but data are limited from human subjects. Dynamic IL13 levels have been demonstrated in mice models in relation to the time period after the incidence of the MI. The levels significantly increased and reached the highest levels on the third day and, after that, started declining until day seven, and then further increased [11]. However, other reports showed an upregulation of IL13 after the seventh day and continued until day fourteen [12]. In the present study, we observed about four folds of elevated IL13 in MI patients compared to healthy controls. Most of the patients enrolled in the present study just after the incidence of MI (within six hours), possibly that could be the reason for higher levels of IL13 in the plasma levels. Further, it has been well documented that elevated IL13 is essential for the cardio-protective effect [11].

There have been limited investigations into the role of IL13 common variants in the development of MI. The homozygous mutant of the rs20541 polymorphism has been linked to MI susceptibility in Greek Cypriot males [25]. In contrast, the Iranian population rs20541 failed to show such an association [24], which is consistent with the findings of the current study. In the Iranian population [24] and in the current study, heterozygotes of IL13 rs1881457 polymorphisms are associated with protection against MI development. Furthermore, the other polymorphism, rs1800925, demonstrated the protective nature of heterozygotes, which were highly prevalent in healthy controls. The precise mechanism by which the variant protects against the development of MI is unknown. Both IL13 (rs1800925 and rs1881457) polymorphisms are located in the intron region. Although these regions are typically not translated into proteins but can affect gene expression by altering splicing patterns, transcription factor binding, or RNA stability. Intronic variants can lead to alternative splicing, exon skipping, or the inclusion of aberrant exons, which may disrupt protein function and contribute to disease [30–32]. The two IL13 polymorphisms (rs1800925 and rs1881457) were possibly linked to plasma levels of IL13, and the intermediate production of IL13 cytokines may protect against the development of MI. The current study has several advantages over previous studies, including i) a larger number of participants and ii) genotyping and quantification of plasma IL13 in each sample.

HWE provides a theoretical framework for predicting the expected frequencies of genotypes in a large, randomly mating population where no evolutionary forces are at play. Any departure from this equilibrium can signal the presence of evolutionary forces, such as natural selection, mutation, migration, genetic drift, or non-random mating. We found a deviation of

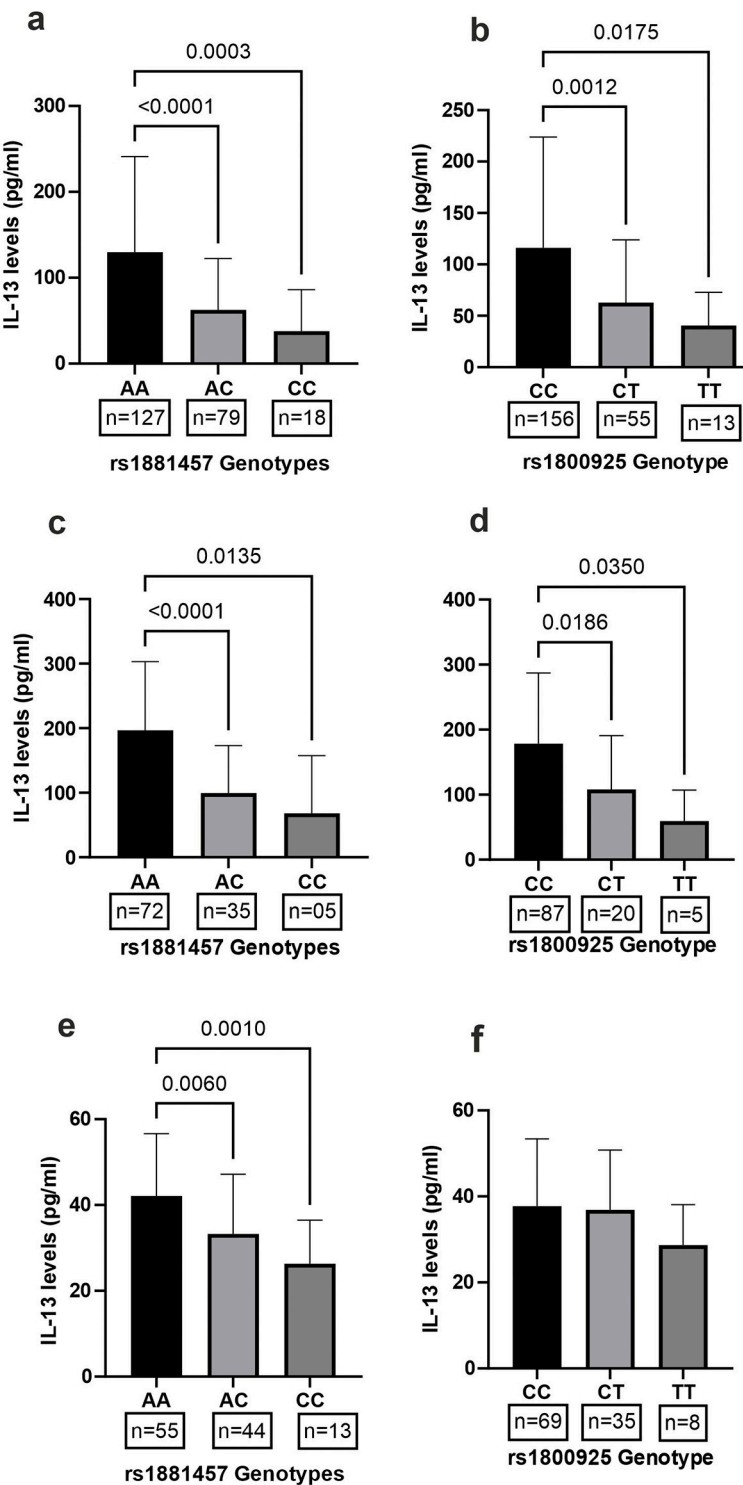

**Fig 2. Association of plasma IL-13 with variants of IL-13 polymorphisms (rs1881457 and rs1800925).** Plasma levels of IL13 were quantified in MI patients (n = 112) and healthy controls (n = 112). Mean IL13 levels in different genotypes of rs1881457 (a) and rs1800925 (b) in the pooled clinical categories (MI patients and healthy controls) were compared by ANOVA. Further, an association of rs1881457 and rs1800925 polymorphism with IL13 was explored in MI patients (c and d) and healthy controls (e and f). A P value <0.05 is taken as statistically significant.

IL13 rs20541 polymorphism genotypes from HWE in this study. HWE deviation has been linked to factors such as population stratification, genotyping error, and selection pressure [33]. We recruited healthy controls from a similar ethnic group and geographical area, reducing the possibility of population stratification. Using stringent genotyping technology, we eliminated the possibility of error in the genotyping method. The deviation could be attributed to the selection pressure provided by different infections [34–37]. Importantly, IL13 rs20541 polymorphism has been linked with susceptibility to pulmonary tuberculosis [38], *Schistosoma mansoni* infection [39], hand, foot, and mouth diseases [40], further strengthening its possibility of the beneficial selection of genetic variants in the studied population.

The current study has revealed a significant role played by IL13 in the pathogenesis of MI. The genetic variations within the IL13 gene, specifically rs1881457 and rs180092, were observed to have varying levels of plasma IL13 and were found to offer protection against the development of MI. Although the precise mechanism linking the genetic mutants to plasma levels of IL13 is not yet fully understood, understanding the potential mechanism could provide valuable information on the underlying mechanisms of atherosclerosis and plaque instability, which are key factors contributing to MI. Identifying specific IL-13 polymorphisms associated with increased MI risk or severity could aid in risk stratification and the early detection of at-risk individuals. Furthermore, IL-13 genetic profiling may help personalize treatment strategies, such as anti-inflammatory therapies or targeted interventions to modulate immune responses, to improve outcomes and prevent recurrent cardiovascular events. Overall, the study of IL-13 gene polymorphism holds promise for enhancing risk assessment, prognosis prediction, and therapeutic interventions in the context of MI and cardiovascular disease management. Although the current study successfully demonstrated the significance of the IL13 genetic variant in MI, it is essential to acknowledge the limitations of the investigation. One limitation is that the Chinese population is a mixed group comprised of various ethnic and sub-population groups. The sample size used in the study may not be sufficient, which warrants further investigation with a larger sample size. Additionally, the study only examined three SNPs in the IL13 gene, so the role of other variants in the predisposition to MI remains unknown. Moreover, the current case-control study did not explore the mechanism of how the genetic variations in the IL13 gene alter the expression of mRNA or respective protein levels. Furthermore, the frequency distribution of IL13 rs20541 genotypes did not conform to the expectations of the HWE, thus rendering the interpretation of the relationship between haplotype and MI cautious.

Based on the observations of this study, future research should concentrate on elucidating the molecular mechanisms underlying the protective effect of IL13 genetic mutants and exploring how IL13 variants modulate gene expression and inflammatory pathways in cardiovascular tissues. Employing transgenic animal models and cellular systems can provide deeper insights into the functional impact of these polymorphisms. Large-scale epidemiological studies are necessary to confirm these findings across diverse populations and understand the gene-environment interactions that influence MI risk. Furthermore, investigating the potential of IL13 polymorphisms as biomarkers for risk stratification and personalized medicine could significantly impact cardiovascular care, leading to targeted prevention and therapeutic strategies.

## Conclusions

The IL13 gene variants rs1881457 and rs180092 have been linked to varying plasma levels of IL13 and offer protection against the development of MI in the Chinese population. This finding adds to the existing body of research on genetic factors influencing MI risk, and it sheds

light on the possible role of IL13 in cardiovascular health. To better comprehend the importance and applicability of these gene variants, further investigations are needed in diverse populations, including those with different ethnic backgrounds. These studies are crucial for validating the findings and determining the broader relevance of IL13 variants in the global context of MI prevention and treatment.

## Acknowledgments

Authors would like to thanks all participants of the study for their voluntary contribution and involvement in the research.

## Author Contributions

**Conceptualization:** Xiaofeng Ma.

**Formal analysis:** Rong Chen, Qiaoling Bao.

**Investigation:** Qiaoling Bao.

**Methodology:** Rong Chen.

**Resources:** Rong Chen.

**Supervision:** Xiaofeng Ma.

**Writing – original draft:** Rong Chen, Qiaoling Bao.

**Writing – review & editing:** Xiaofeng Ma.

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
