## [Decision Letter · Decision Letter 0]

28 May 2024

PONE-D-24-15689Association of IL13 polymorphisms with susceptibility to myocardial infarction: a case-control study in Chinese population.PLOS ONE

Dear Dr. Ma,

Thank you for submitting your manuscript to PLOS ONE. After careful consideration, we feel that it has merit but does not fully meet PLOS ONE’s publication criteria as it currently stands. Therefore, we invite you to submit a revised version of the manuscript that addresses the points raised during the review process.

We look forward to receiving your revised manuscript.

Kind regards,

Andrea Da Porto

Academic Editor

PLOS ONE

Journal Requirements:

2. In the online submission form, you indicated that [All data are given in the manuscript. if further info is required, may be requested to the corresponding author.]. 

Additional Editor Comments:

Dear Authors,

Your manuscript has been reviewed by experts in the field and we request that you make major revisions before it is processed further. Please address all the reviewers comment that you find below.

Please revise the manuscript according to the reviewers' comments and upload the revised file within 7 days.

Reviewers comments

Reviewer One

Revision Comments for the Authors

Dear Authors,

Thank you for submitting your manuscript titled "Association of IL13 Polymorphisms with Susceptibility to Myocardial Infarction: A Case-Control Study in Chinese Population" to PLOS ONE. Your study presents important findings regarding the genetic factors contributing to myocardial infarction (MI) susceptibility. After careful review, I have some comments and suggestions that could improve the clarity and rigor of your manuscript.

Clarity and Ambiguity

1. Introduction:

- The introduction provides a good overview but could benefit from more concise language and clearer transitions between topics. For instance, the shift from discussing MI prevalence to genetic factors could be smoother.

- Clearly state the hypothesis early in the introduction to guide the reader through the rationale of the study.

2. Materials and Methods:

- Some details are missing or could be expanded for better clarity. For example, the criteria for selecting controls need to be more detailed. Explicitly mention the criteria for excluding subjects (e.g., based on BP, cholesterol, and sugar levels). - Describe the process of informed consent in more detail. This helps in understanding how ethical guidelines were followed.

3. Genotyping and ELISA Methods:

- The description of the genotyping and ELISA methods could be more detailed to ensure reproducibility. Include information about the conditions used in the TaqMan Genotyping assays and ELISA procedures.

- Clearly mention the sources and catalog numbers of the kits used for genotyping and ELISA.

4. Statistical Analysis:

- The statistical methods section should be more detailed. Explain the choice of tests (e.g., why Fisher's exact test was used) and how the Bonferroni correction was applied.

- Clarify the use of software tools, including versions and any specific settings or parameters used in the analyses.

5. Results:

- Results are presented clearly, but some sections could benefit from additional context or explanation. For example, explain the implications of the Hardy-Weinberg equilibrium results in lay terms.

- Ensure that all tables and figures are referenced in the text and include legends that are descriptive enough to be understood independently of the main text.

6. Discussion:

- The discussion is comprehensive but can be more concise. Some points are repeated or could be merged for better flow. - Discuss the limitations more explicitly and suggest future research directions based on your findings.

7. Conclusion:

- The conclusion is clear but could briefly restate the significance of the findings in the broader context of MI research.

Specific Areas of Ambiguity

1. HWE Deviation:

- The deviation from HWE for rs20541 is mentioned, but the reasons and implications are not fully explained. This could be confusing for readers not familiar with genetic studies.

2. IL-13 Levels Timing:

- It is mentioned that most patients were enrolled soon after MI, but the timing of IL-13 level measurements relative to MI onset should be more precisely described.

3. Terminology:

- Ensure that all technical terms are defined when first used. For example, "Th2 immune response" and "pleiotropic effect" might need brief explanations.

Additional Comments:

1. Sample Size Calculation and Statistical Power:

- Please include a detailed explanation of how the sample size was determined. Indicate whether a priori power analysis was conducted to ensure that the study is adequately powered to detect significant associations.

- Specify the effect size, significance level, and power used in the sample size calculation.

2. Abbreviations:

- Ensure that all abbreviations are defined at their first mention in the text. For example, abbreviations such as MI (myocardial infarction), SNP (single nucleotide polymorphism), and HWE (Hardy-Weinberg equilibrium) should be defined upon first use.

- Provide a list of abbreviations either at the beginning or end of the manuscript for reference.

3. Demographic and Clinical Characteristics:

- Provide more detailed demographic and clinical characteristics of the study participants, including information on smoking status, family history of cardiovascular diseases, and other relevant factors.

4. Genotyping Quality Control:

- Describe the quality control measures taken during genotyping to ensure accuracy and reliability of the data. This includes the handling of potential genotyping errors and missing data.

- Mention if any duplicate samples or independent replication was performed to validate the genotyping results.

5. Hardy-Weinberg Equilibrium (HWE):

- Discuss the deviation from HWE observed for the rs20541 polymorphism in more detail. Explain potential reasons for this deviation and how it might affect the study's conclusions.

- Consider conducting sensitivity analyses to determine if the HWE deviation impacts the association results.

6. Statistical Analysis:

- Provide more details on the statistical methods used for haplotype analysis and how the haplotype frequencies were compared between cases and controls.

- Clarify how the Bonferroni correction was applied and ensure that all reported p-values reflect this correction where multiple comparisons are made.

7. Plasma IL13 Quantification:

- Discuss whether IL13 levels were measured at a consistent time point post-MI for all patients to account for the dynamic changes in IL13 levels after MI.

8. Discussion and Interpretation:

- Expand on the biological plausibility of the associations found, particularly how the identified polymorphisms might influence IL13 expression and MI risk.

- Discuss the potential clinical implications of your findings and any limitations that should be considered when interpreting the results.

9. Ethical Approval:

- Include the specific ethical committee approval code for the study. This is crucial for ensuring transparency and adherence to ethical standards in research.

10. Supplementary Information:

- Provide any supplementary information or detailed protocols as supporting documents to allow for replication of your study by other researchers.

By addressing these points, you will enhance the robustness and transparency of your study. I look forward to seeing the revised version of your manuscript.

Best regards,

REVIEWER 2

The present study was a candidate gene association study for the association of IL-13 polymorphism with susceptibility to myocardial infarction (MI) in a Chinese population. Although the topic was good, there were some concerns and errors that should be addressed and revised by the authors.

Major:

1- Chinese population consists of many ethnicities and sub-populations. The sample size is not large enough for such a mixed population.

2- The effect of the studied polymorphism in genome-wide association studies (GWAS) should be mentioned in the state of the problem. Then it should be clarified whether this candidate gene association study is for confirmation of a GWAS result or is logic based unrelated to GWAS.

3- Material and methods should be started with a subsection entitled "study design". In this subsection, STREGA guideline should be mentioned.

4- The manuscript should be updated based on STREGA guideline.

5- Although TaqMan genotyping is an advanced method, I could not find any information regarding its related PCR.

6- The outcome and its measurement should be defined in material and methods.

7- I could not find study limitations in the discussion.

Minor:

1- All manufacturers and software packages should be addressed as (Company, State, Country).

2- "A P value less than 0.05 was considered statistically significant", it should be clarified that this sentence is for which test as you used Bonferroni correction for testing genotypes.

3- Table 2, why VDR?

Reviewers' comments:

Reviewer's Responses to Questions

**Comments to the Author**

1. Is the manuscript technically sound, and do the data support the conclusions?

Reviewer #1: Yes

Reviewer #2: Yes

Reviewer #3: Yes

Reviewer #4: Partly

2. Has the statistical analysis been performed appropriately and rigorously? 

Reviewer #1: Yes

Reviewer #2: Yes

Reviewer #3: Yes

Reviewer #4: No

3. Have the authors made all data underlying the findings in their manuscript fully available?

Reviewer #1: Yes

Reviewer #2: Yes

Reviewer #3: Yes

Reviewer #4: No

4. Is the manuscript presented in an intelligible fashion and written in standard English?

Reviewer #1: Yes

Reviewer #2: Yes

Reviewer #3: Yes

Reviewer #4: No

5. Review Comments to the Author

Reviewer #1: Revision Comments for the Authors

Dear Authors,

Thank you for submitting your manuscript titled "Association of IL13 Polymorphisms with Susceptibility to Myocardial Infarction: A Case-Control Study in Chinese Population" to PLOS ONE. Your study presents important findings regarding the genetic factors contributing to myocardial infarction (MI) susceptibility. After careful review, I have some comments and suggestions that could improve the clarity and rigor of your manuscript.

Clarity and Ambiguity

1. Introduction:

- The introduction provides a good overview but could benefit from more concise language and clearer transitions between topics. For instance, the shift from discussing MI prevalence to genetic factors could be smoother.

- Clearly state the hypothesis early in the introduction to guide the reader through the rationale of the study.

2. Materials and Methods:

- Some details are missing or could be expanded for better clarity. For example, the criteria for selecting controls need to be more detailed. Explicitly mention the criteria for excluding subjects (e.g., based on BP, cholesterol, and sugar levels). - Describe the process of informed consent in more detail. This helps in understanding how ethical guidelines were followed.

3. Genotyping and ELISA Methods:

- The description of the genotyping and ELISA methods could be more detailed to ensure reproducibility. Include information about the conditions used in the TaqMan Genotyping assays and ELISA procedures.

- Clearly mention the sources and catalog numbers of the kits used for genotyping and ELISA.

4. Statistical Analysis:

- The statistical methods section should be more detailed. Explain the choice of tests (e.g., why Fisher's exact test was used) and how the Bonferroni correction was applied.

- Clarify the use of software tools, including versions and any specific settings or parameters used in the analyses.

5. Results:

- Results are presented clearly, but some sections could benefit from additional context or explanation. For example, explain the implications of the Hardy-Weinberg equilibrium results in lay terms.

- Ensure that all tables and figures are referenced in the text and include legends that are descriptive enough to be understood independently of the main text.

6. Discussion:

- The discussion is comprehensive but can be more concise. Some points are repeated or could be merged for better flow. - Discuss the limitations more explicitly and suggest future research directions based on your findings.

7. Conclusion:

- The conclusion is clear but could briefly restate the significance of the findings in the broader context of MI research.

Specific Areas of Ambiguity

1. HWE Deviation:

- The deviation from HWE for rs20541 is mentioned, but the reasons and implications are not fully explained. This could be confusing for readers not familiar with genetic studies.

2. IL-13 Levels Timing:

- It is mentioned that most patients were enrolled soon after MI, but the timing of IL-13 level measurements relative to MI onset should be more precisely described.

3. Terminology:

- Ensure that all technical terms are defined when first used. For example, "Th2 immune response" and "pleiotropic effect" might need brief explanations.

Additional Comments:

1. Sample Size Calculation and Statistical Power:

- Please include a detailed explanation of how the sample size was determined. Indicate whether a priori power analysis was conducted to ensure that the study is adequately powered to detect significant associations.

- Specify the effect size, significance level, and power used in the sample size calculation.

2. Abbreviations:

- Ensure that all abbreviations are defined at their first mention in the text. For example, abbreviations such as MI (myocardial infarction), SNP (single nucleotide polymorphism), and HWE (Hardy-Weinberg equilibrium) should be defined upon first use.

- Provide a list of abbreviations either at the beginning or end of the manuscript for reference.

3. Demographic and Clinical Characteristics:

- Provide more detailed demographic and clinical characteristics of the study participants, including information on smoking status, family history of cardiovascular diseases, and other relevant factors.

4. Genotyping Quality Control:

- Describe the quality control measures taken during genotyping to ensure accuracy and reliability of the data. This includes the handling of potential genotyping errors and missing data.

- Mention if any duplicate samples or independent replication was performed to validate the genotyping results.

5. Hardy-Weinberg Equilibrium (HWE):

- Discuss the deviation from HWE observed for the rs20541 polymorphism in more detail. Explain potential reasons for this deviation and how it might affect the study's conclusions.

- Consider conducting sensitivity analyses to determine if the HWE deviation impacts the association results.

6. Statistical Analysis:

- Provide more details on the statistical methods used for haplotype analysis and how the haplotype frequencies were compared between cases and controls.

- Clarify how the Bonferroni correction was applied and ensure that all reported p-values reflect this correction where multiple comparisons are made.

7. Plasma IL13 Quantification:

- Discuss whether IL13 levels were measured at a consistent time point post-MI for all patients to account for the dynamic changes in IL13 levels after MI.

8. Discussion and Interpretation:

- Expand on the biological plausibility of the associations found, particularly how the identified polymorphisms might influence IL13 expression and MI risk.

- Discuss the potential clinical implications of your findings and any limitations that should be considered when interpreting the results.

9. Ethical Approval:

- Include the specific ethical committee approval code for the study. This is crucial for ensuring transparency and adherence to ethical standards in research.

10. Supplementary Information:

- Provide any supplementary information or detailed protocols as supporting documents to allow for replication of your study by other researchers.

By addressing these points, you will enhance the robustness and transparency of your study. I look forward to seeing the revised version of your manuscript.

Best regards,

Reviewer #2: The present study was a candidate gene association study for the association of IL-13 polymorphism with susceptibility to myocardial infarction (MI) in a Chinese population. Although the topic was good, there were some concerns and errors that should be addressed and revised by the authors.

Major:

1- Chinese population consists of many ethnicities and sub-populations. The sample size is not large enough for such a mixed population.

2- The effect of the studied polymorphism in genome-wide association studies (GWAS) should be mentioned in the state of the problem. Then it should be clarified whether this candidate gene association study is for confirmation of a GWAS result or is logic based unrelated to GWAS.

3- Material and methods should be started with a subsection entitled "study design". In this subsection, STREGA guideline should be mentioned.

4- The manuscript should be updated based on STREGA guideline.

5- Although TaqMan genotyping is an advanced method, I could not find any information regarding its related PCR.

6- The outcome and its measurement should be defined in material and methods.

7- I could not find study limitations in the discussion.

Minor:

1- All manufacturers and software packages should be addressed as (Company, State, Country).

2- "A P value less than 0.05 was considered statistically significant", it should be clarified that this sentence is for which test as you used Bonferroni correction for testing genotypes.

3- Table 2, why VDR?!

Reviewer #3: After reading the paper i have come to the following conclusions:

1-This paper is Technically sound, all the steps were explained clearly and concisely in a manner that allows replication by future researchers. The limitations of the paper were also addressed as well as all supporting data in clear and chronological order.

2-Statistical analysis supported findings, with mention of methods, software and results. All results were easy to understand and thoroughly explained.

3-It was easy to read through this paper, it was well written, easy to comprehend, andI am confident that those seeking knowledge from different backgrounds will be able to reach similar conclusions to the authors with ease.

This s a job well done!

Reviewer #4: -Correlation between risk factors and IL 13 polymorphism as DM , Dyslipidemia and hypertension are not mentioned.

-Type of myocardial infarction NSTEMI, STEMI is not clear.

-Detailed data about the lipid profile (HDL,LDL TG and total cholesterol level are beneficial.

-Many studies are previously published with the same ideas and population thus this study add no much benefits .

6. PLOS authors have the option to publish the peer review history of their article (what does this mean?). If published, this will include your full peer review and any attached files.

Reviewer #1: No

Reviewer #2: **Yes: **Seyyed Amir Yasin Ahmadi

Reviewer #3: **Yes: **Maha Ahmed

Reviewer #4: No

---

## [Author Response · Author response to Decision Letter 0]

25 Jun 2024

Additional Editor Comments:

Dear Authors,

Your manuscript has been reviewed by experts in the field and we request that you make major revisions before it is processed further. Please address all the reviewers comment that you find below.

Please revise the manuscript according to the reviewers' comments and upload the revised file within 7 days.

Dear Editor, Thanks for your mail and for allowing us to revise our manuscript. As I was not well and fit, and was away from the office, I couldn’t revise the manuscript within 7 days. I sincerely apologise for that. Further, I wish to thank you for acting as an editor and obtaining comments from all potent reviewers. We learned several things from the reviewer's comments and revised the manuscript accordingly. We strongly believe the manuscript has improved significantly compared to the first version. 

With regards

Reviewers comment

Reviewer One

Revision Comments for the Authors

Dear Authors,

Thank you for submitting your manuscript titled "Association of IL13 Polymorphisms with Susceptibility to Myocardial Infarction: A Case-Control Study in Chinese Population" to PLOS ONE. Your study presents important findings regarding the genetic factors contributing to myocardial infarction (MI) susceptibility. After careful review, I have some comments and suggestions that could improve the clarity and rigor of your manuscript.

Clarity and Ambiguity

1. Introduction:

- The introduction provides a good overview but could benefit from more concise language and clearer transitions between topics. For instance, the shift from discussing MI prevalence to genetic factors could be smoother.

Answer: We agree with the esteemed reviewer's comment. For a smoother transition, an additional 4-5 lines have been incorporated between the paragraphs. We have also merged both paragraphs in the revised version of the manuscript. (Page no: 3 , Paragraph no: 1 )

- Clearly state the hypothesis early in the introduction to guide the reader through the rationale of the study.

Answer: We maintained a write-up flow from the prevalence of MI, the role of cytokines, how genetic variants alter the cytokines levels, and their association with the predisposition to MI, and in the last paragraph, we proposed our hypothesis. We believe putting the hypothesis at the end of the introduction section is ideal for the introduction to the reader. 

2. Materials and Methods:

- Some details are missing or could be expanded for better clarity. For example, the criteria for selecting controls need to be more detailed. Explicitly mention the criteria for excluding subjects (e.g., based on BP, cholesterol, and sugar levels). - Describe the process of informed consent in more detail. This helps in understanding how ethical guidelines were followed.

Answer: We completely agree with the reviewer's observations. The revised manuscript includes details of the healthy controls. In addition, the informed consent procedure is spelt out in detail. (Page no: 5 , last Paragraph)

3. Genotyping and ELISA Methods:

- The description of the genotyping and ELISA methods could be more detailed to ensure reproducibility. Include information about the conditions used in the TaqMan Genotyping assays and ELISA procedures.

Answer: We agree. Details of the ELISA and TaqMan Genotyping assays has been incorporated in the revised manuscript briefly to ensure the reproducibility of the methods. (Page no: 6 and 7 , Paragraph no: last and second respectively)

- Clearly mention the sources and catalog numbers of the kits used for genotyping and ELISA.

Answer: We agree. The catalogue number of ELISA and TaqMan assay kit has been included in the revised manuscript. (Page no: 6 and 7 , Paragraph no: last and second respectively)

4. Statistical Analysis:

- The statistical methods section should be more detailed. Explain the choice of tests (e.g., why Fisher's exact test was used) and how the Bonferroni correction was applied.

Answer: We agree. Details of the Fisher exact test and Bonferroni correction with appropriate references have been incorporated in the statistical analysis sub-section of the revised manuscript. (Page no: 8, Paragraph no: 1)

- Clarify the use of software tools, including versions and any specific settings or parameters used in the analyses.

Answer: We agree. Details of the software and settings have been included in the manuscript.

5. Results:

- Results are presented clearly, but some sections could benefit from additional context or explanation. For example, explain the implications of the Hardy-Weinberg equilibrium results in lay terms.

Answer: We agree. Using lay terms, we have included one paragraph on Hardy-Weinberg equilibrium in the result section of the revised manuscript. (Page no: 10 and 11, Paragraph no: last and first)

- Ensure that all tables and figures are referenced in the text and include legends that are descriptive enough to be understood independently of the main text.

Answer: We agree. We checked all tables and figures and ensure that all are cited in the text. Further, we improved the legends of figures and footnotes of the Table wherever required. 

6. Discussion:

- The discussion is comprehensive but can be more concise. Some points are repeated or could be merged for better flow. - Discuss the limitations more explicitly and suggest future research directions based on your findings.

Answer: We agree. The discussion portion has been revised. In addition, both limitations of the present study and future research directions were mentioned in the discussion subsection of the revised manuscript. (Page no: 16, Paragraph no: 2)

7. Conclusion:

- The conclusion is clear but could briefly restate the significance of the findings in the broader context of MI research.

Answer: We agree. The conclusion has been rewritten in the revised manuscript. (Page no: 17, Paragraph no: 2)

Specific Areas of Ambiguity

1. HWE Deviation:

- The deviation from HWE for rs20541 is mentioned, but the reasons and implications are not fully explained. This could be confusing for readers not familiar with genetic studies.

Answer: We agree. The revised manuscript elaborately demonstrates the implications of the HWE-deviated genotype. (Page no: 15, last Paragraph)

2. IL-13 Levels Timing:

- It is mentioned that most patients were enrolled soon after MI, but the timing of IL-13 level measurements relative to MI onset should be more precisely described.

Answer: We agree. For most of the patients the blood sample was collected within 6 hours of onset of MI. This statement has been incorporated in the revised manuscript. (Page no: 6, first Paragraph)

3. Terminology:

- Ensure that all technical terms are defined when first used. For example, "Th2 immune response" and "pleiotropic effect" might need brief explanations.

Answer: We agree. We have defined the statements in the revised manuscript. (Page no: 3) 

Additional Comments:

1. Sample Size Calculation and Statistical Power:

- Please include a detailed explanation of how the sample size was determined. Indicate whether a priori power analysis was conducted to ensure that the study is adequately powered to detect significant associations.

Answer: A priori power calculation was performed to detect a significant association with the enrolment of an adequate number of samples. Details are mentioned in a subheading of the materials and methods section. (Page no: 6, Paragraph no: 2)

- Specify the effect size, significance level, and power used in the sample size calculation.

Answer: We agree. Details of effect size, significance levels, and power used to calculate sample size have been incorporated in the revised manuscript under the Materials and Methods section. (Page no: 6, Paragraph no: 2)

2. Abbreviations:

- Ensure that all abbreviations are defined at their first mention in the text. For example, abbreviations such as MI (myocardial infarction), SNP (single nucleotide polymorphism), and HWE (Hardy-Weinberg equilibrium) should be defined upon first use.

Answer: We agree. We have checked throughout the manuscript and corrected the abbreviations as needed.

- Provide a list of abbreviations either at the beginning or end of the manuscript for reference.

Answer: We agree. A list of abbreviations has been included at the end of the manuscript. (Page no: 20)

3. Demographic and Clinical Characteristics:

- Provide more detailed demographic and clinical characteristics of the study participants, including information on smoking status, family history of cardiovascular diseases, and other relevant factors.

Answer: We agree with the comment. Demographic data, such as the family history of cardiovascular diseases, smoking, and excessive alcohol consumption, were included in the baseline table and compared among the MI cases and healthy controls. In addition, the lipid profile and cholesterol levels are also included in the revised manuscript (Table-1)

4. Genotyping Quality Control:

- Describe the quality control measures taken during genotyping to ensure accuracy and reliability of the data. This includes the handling of potential genotyping errors and missing data.

Answer: We agree. The genotyping quality control and handling of missing data have been included in the revised manuscript. Further, we have mentioned the actual sample number (MI: 328 and HC: 322), and after successful genotyping, only 310 cases and 305 healthy controls were enrolled in the study.” (Page no: 7, Paragraph no: 1)

- Mention if any duplicate samples or independent replication was performed to validate the genotyping results.

Answer: Yes, we have taken samples on duplicate to validate the genotyping results. This has been mentioned in the revised manuscript. (Page no: 7, Paragraph no: 1)

5. Hardy-Weinberg Equilibrium (HWE):

- Discuss the deviation from HWE observed for the rs20541 polymorphism in more detail. Explain potential reasons for this deviation and how it might affect the study's conclusions.

Answer: In the discussion section, we have already described the deviation of rs20541 from HWE. Further, we have included 3-4 on basic HWE in the discussion section. The possible reason for the deviation from HWE has already been described. The deviation of rs20441 genotype distribution from HWE won't affect the conclusion of the study as the genetic associations between IL13 polymorphism and MI were observed for the other two SNPs. However, it may affect the haplotype analysis. We have included these statements in the limitation subsection of the manuscript. (Page no: 15, last Paragraph)

- Consider conducting sensitivity analyses to determine if the HWE deviation impacts the association results.

Answer: We agree. In the present investigation, the genotypes of the rs20541 polymorphism were examined in individuals with MI and healthy controls. Moreover, the distribution of alleles and genotypes for the rs20541 polymorphism was compared between the MI patients and healthy controls, revealing no significant association between the polymorphism and the predisposition to MI in the studied cohort. We are in the process of planning a meta-analysis to assess the association of the IL13 polymorphism with a predisposition to MI. Additionally, we will conduct a sensitivity analysis to determine whether any deviation from HWE impacts the association results. 

6. Statistical Analysis:

- Provide more details on the statistical methods used for haplotype analysis and how the haplotype frequencies were compared between cases and controls.

Answer: We agree. Details of haplotype comparison have been provided in the statistical analysis sub-section of materials and methods. (Page no: 7, Paragraph no:1 )

- Clarify how the Bonferroni correction was applied and ensure that all reported p-values reflect this correction where multiple comparisons are made.

Answer: we agree. The Bonferroni corrections were performed for the genotype, allele and haplotype comparison. Details are mentioned in the statistical analysis subsection of the revised manuscript. (Page no: 7, Paragraph no: 1)

7. Plasma IL13 Quantification:

- Discuss whether IL13 levels were measured at a consistent time point post-MI for all patients to account for the dynamic changes in IL13 levels after MI.

Answer: We agree. The levels of IL13 dynamic changes to time. To maintain uniformity, we only considered patients and collected within 6 hours of the onset of MI. This has been mentioned in the method section of the revised manuscript. (Page no: 6, Paragraph no:1 )

8. Discussion and Interpretation:

- Expand on the biological plausibility of the associations found, particularly how the identified polymorphisms might influence IL13 expression and MI risk.

Answer: We agree. We have included the possible biological associations of gene polymorphisms with an expression of IL13 with appropriate references. (Page no: 15, Paragraph no:2 )

- Discuss the potential clinical implications of your findings and any limitations that should be considered when interpreting the results.

Answer: We agree. The potential clinical implications of the study and limitations are included in the discussion section of the revised manuscript. (Page no: 16, last Paragraph)

9. Ethical Approval:

- Include the specific ethical committee approval code for the study. This is crucial for ensuring transparency and adherence to ethical standards in research.

Answer: We accept. The ethical committee approval code for the study has been incorporated into the manuscript. Additionally, details regarding the consent process have been added to the revised paper. (Page no: 5, Paragraph no: 1)

10. Supplementary Information:

- Provide any supplementary information or detailed protocols as supporting documents to allow for replication of your study by other researchers.

Answer: The Materials and Methods section of the revised manuscript has been thoroughly revised to include a comprehensive and detailed protocol that aims to facilitate the replication of the method by future researchers. It should be noted that no supplementary information regarding the methods has been included in the revised manuscript.

By addressing these points, you will enhance the robustness and transparency of your study. I look forward to seeing the revised version of your manuscript.

Best regards,

Answer: We extend our sincere gratitude to the anonymous reviewer for providing invaluable feedback that significantly enhanced our manuscript. We firmly believe that the revised version of our paper has been greatly improved, and we express our profound appreciation to the reviewer. 

REVIEWER 2

The present study was a candidate gene association study for the association of IL-13 polymorphism with susceptibility to myocardial infarction (MI) in a Chinese population. Although the topic was good, there were some concerns and errors that should be addressed and revised by the authors.

Major:

1- Chinese population consists of many ethnicities and sub-populations. The sample size is not large enough for such a mixed population.

Answer: We agree with the esteemed reviewer's observations, which are mentioned in the study's limitations. (Page no: 16, last Paragraph)

2- The effect of the studied polymorphism in genome-wide association studies (GWAS) should be mentioned in the state of the problem. Then it should be clarified whether this candidate gene association study is for confirmation of a GWAS result or is logic based unrelated to GWAS.

Answer: We agree. In the introduction section 4-5 lines have been incorporated. None of the GWAS have shown the association of IL13 with MI susceptibility. Thus, the present investigation was undertaken based on the differential IL13 cytokine levels in MI patients compared to HC and different clinical manifestations. (Page no: 4, Paragraph no:2 )

3- Material and methods should be started with a subsection entitled "study design". In this subsection, STREGA guideline should be mentioned.

Answer: We agree. We have included a subsection named study design in the Materials and Methods and disclosed adherence to STREGA guidelines while reporting the study. (Page no: 5, Paragraph no:3 )

4- The manuscript should be updated based on STREGA guideline.

Answer: The whole manuscript 

---

## [Decision Letter · Decision Letter 1]

17 Jul 2024

Association of IL13 polymorphisms with susceptibility to myocardial infarction: a case-control study in Chinese population.

PONE-D-24-15689R1

Dear Dr. Xiaofeng Ma,

We’re pleased to inform you that your manuscript has been judged scientifically suitable for publication and will be formally accepted for publication once it meets all outstanding technical requirements.

Kind regards,

Andrea Da Porto

Academic Editor

PLOS ONE

Additional Editor Comments (optional):

Reviewers' comments:

Reviewer's Responses to Questions

**Comments to the Author**

1. If the authors have adequately addressed your comments raised in a previous round of review and you feel that this manuscript is now acceptable for publication, you may indicate that here to bypass the “Comments to the Author” section, enter your conflict of interest statement in the “Confidential to Editor” section, and submit your "Accept" recommendation.

Reviewer #1: All comments have been addressed

Reviewer #2: All comments have been addressed

Reviewer #3: All comments have been addressed

2. Is the manuscript technically sound, and do the data support the conclusions?

Reviewer #1: Yes

Reviewer #2: Yes

Reviewer #3: Yes

3. Has the statistical analysis been performed appropriately and rigorously? 

Reviewer #1: Yes

Reviewer #2: Yes

Reviewer #3: Yes

4. Have the authors made all data underlying the findings in their manuscript fully available?

Reviewer #1: Yes

Reviewer #2: (No Response)

Reviewer #3: Yes

5. Is the manuscript presented in an intelligible fashion and written in standard English?

Reviewer #1: Yes

Reviewer #2: Yes

Reviewer #3: Yes

6. Review Comments to the Author

Reviewer #1: Dear Authors, Thank you for submitting your manuscript. Your work presents a thorough analysis and provides significant insights into the subject matter.

Best regards

Reviewer #2: My comments except Major 1 were fully resolved by the author. Acceptance of this manuscript depends on editorial decision about my comment Major 1.

Reviewer #3: The previous draft was satisfactory, but after answering all the reviewers comments ,this paper

transcended to even better levels.

I'm exited to see it replicated in other ethnic groups and maybe one day in a meta analysis.

7. PLOS authors have the option to publish the peer review history of their article (what does this mean?). If published, this will include your full peer review and any attached files.

Reviewer #1: No

Reviewer #2: **Yes: **Seyyed Amir Yasin Ahmadi

Reviewer #3: **Yes: **Maha Ahmed

---

## [Editor Report · Acceptance letter]

24 Jul 2024

PONE-D-24-15689R1 

PLOS ONE

Dear Dr. Ma, 

I'm pleased to inform you that your manuscript has been deemed suitable for publication in PLOS ONE. Congratulations! Your manuscript is now being handed over to our production team.

Kind regards, 

on behalf of

Dr. Andrea Da Porto 

Academic Editor

PLOS ONE